

# Improved GOMOS/Envisat ozone retrievals in the upper troposphere and the lower stratosphere

Viktoria F. Sofieva[1], Iolanda Ialongo[1], Janne Hakkarainen[1], Erkki Kyrölä[1], Johanna Tamminen[1], Marko Laine[1], Alain Hauchecorne[2], Francis Dalaudier[2], Jean-Loup Bertaux[2], Didier Fussen[3], Laurent Blanot[4], Gilbert Barrot[4], Angelika Dehn[5]

[1] Finnish Meteorological Institute, Helsinki, 00101, Finland
[2] Université Versailles St-Quentin, UPMC University Paris 06, CNRS/INSU, LATMOS-IPSL, 78280 Guyancourt, France
[3] Belgian Institute for Space Aeronomy (IASB-BIRA), Brussels, Belgium
[4] ACRI-ST, Sophia-Antipolis, France
[5] ESA/ESRIN, Frascati, Italy

*Correspondence to*: Viktoria Sofieva (viktoria.sofieva@fmi.fi)

**Abstract.** Global Ozone Monitoring by Occultation of Stars (GOMOS) on board Envisat has performed about 440 000 night-time occultations during 2002–2012. Self-calibrating measurement principle, good vertical resolution, excellent pointing accuracy and the wide vertical range from the troposphere up to the lower thermosphere make GOMOS profiles interesting for different analyses.

The GOMOS ozone data are of high quality in the stratosphere and the mesosphere, but the current operational retrieval algorithm (IPF v.6) is not optimized for retrievals in the upper troposphere–lower stratosphere (UTLS). In particular, validation of GOMOS profiles against ozonesonde data has revealed a substantial positive bias (up to 100%) in the UTLS region. The retrievals in the UTLS are challenging because of low signal-to-noise ratio and the presence of clouds.

In this work, we discuss the reasons for the systematic uncertainties in the UTLS with the IPF v.6 algorithm or its modifications based on simultaneous retrievals of several constituents using the full visible wavelength range. The main reason is high sensitivity of the UTLS retrieval algorithms to an assumed aerosol extinction model.

We have developed a new aerosol–insensitive ozone profile inversion algorithm for GOMOS data in the UTLS using a DOAS-type method at visible wavelengths. The method uses minimal assumptions about the atmospheric profiles. The ozone retrievals in the whole altitude range from the troposphere to the lower thermosphere are performed in two steps, as in the operational algorithm: spectral inversion followed by the vertical inversion. The horizontal column ozone densities retrieved in the spectral inversion follow V6 profiles in the middle atmosphere and follow the triplet ozone profiles in the UTLS. The vertical inversion is performed as in IPF v6 with the Tikhonov-type regularization according to the target resolution.

The validation of new retrieved ozone profiles with ozonesondes show dramatic reduction of GOMOS ozone biases in the UTLS. The new GOMOS ozone profiles are also in a very good agreement with measurements by MIPAS, ACE-FTS and OSIRIS satellite instruments in the UTLS. It is also shown that the known geophysical phenomena in the UTLS ozone are well reproduced with the new GOMOS data.





# 1    Introduction

Monitoring ozone concentration in the UTLS (Upper Troposphere - Lower Stratosphere) is important, because the processes occurring in this region can strongly affect surface climate (Gettelman et al., 2011). Satellite sensors provide ozone profiles with high temporal and spatial resolution, necessary to resolve the strong variability in the UTLS (SPARC, 2000). However,

the UTLS region is difficult for exploration from space. Most nadir-looking instruments do not have a sufficient vertical resolution to resolve strong vertical gradients of interest, while limb-viewing instruments have difficulties because the atmosphere becomes nearly opaque at low altitudes. In addition, the clouds are often sufficiently thick to stop many instruments from seeing through them.

This paper is dedicated to the validation of ozone profiles by the GOMOS (Global Ozone Monitoring by Occultation of

Stars) instrument on board the ENVISAT satellite (2002-2012) with the specific focus on the UTLS region and to the optimization of GOMOS retrievals in the UTLS.

GOMOS is a stellar occultation instrument operating in UV-VIS-NIR wavelength region (Bertaux et al., 2010; Kyrölä et al., 2004). The atmospheric transmission spectra, which are obtained after dividing the stellar spectra observed through the Earth atmosphere by the reference spectrum, recorded above the atmosphere, contain spectral features of absorption and scattering

by gases and particles. Ozone, $NO_2$, $NO_3$, and aerosol extinction are retrieved from the UV-VIS spectrometer data. Ozone can be retrieved up to ~100 km, while other species are detectable in the stratosphere and in the upper troposphere. The lowest altitude of the GOMOS measurements depends on stellar brightness and the presence of clouds (Tamminen et al., 2010); it is usually between 5 and 20 km. The refractive attenuation and perturbations due to scintillation are removed from the GOMOS transmission spectra before the inversion (Kyrölä et al., 2010; Sofieva et al., 2009). The GOMOS data

processing relies on the two-step inversion (Kyrölä et al., 2010). First, atmospheric transmission spectra from every tangent height are inverted to horizontal column densities (along the path of the light beam from the star) for gases and optical thickness for aerosols (spectral inversion). Then, for every constituent, the collection of the horizontal column densities at successive tangent heights is inverted to local density profiles (vertical inversion).

Since the aerosol extinction spectrum is not known a priori, a polynomial of wavelength is used for the description of the

aerosol extinction in the GOMOS retrievals. In the early version of GOMOS processor (Instrument Processor Facility (IPF) v4.02), a simplified aerosol extinction model proportional to $1/\lambda$ ($\lambda$ is wavelength) has been used (Vanhellemont et al., 2005), while further developments (IPF v5.0 and IPF v6.0) used a second-degree polynomial model in $\lambda$ for the description of the aerosol extinction in GOMOS retrievals (Kyrölä et al., 2010; Vanhellemont et al., 2010). The study by Tamminen et al. (2010) has shown that the retrieval of GOMOS ozone profiles in the UTLS is highly sensitive to the aerosol model used.

In particular, using higher order ($2^{nd}$ and $3^{rd}$ order) polynomial aerosol models results in ~30% larger ozone values in the UTLS and in the troposphere compared to the lower order model (zero or $1^{st}$ order).

The vertical resolution (including the smoothing properties of the inversion) of GOMOS ozone profiles is 2 km below 30 km and 3 km above 40 km; it is the same for all occultations (Kyrölä et al., 2010; Sofieva et al., 2004; Tamminen et al., 2010).



The stellar flux recorded by GOMOS, and thus signal-to-noise ratio and uncertainty of retrieved profiles, depends on stellar magnitude and spectral class. The estimated random uncertainty of GOMOS ozone profiles is 0.5-5% in the stratosphere and 1-10% in the mesosphere and lower thermosphere. Validation of the uncertainty estimates for ozone profiles in the stratosphere has shown that they are realistic except for dim stars (Sofieva et al., 2014a).

A dedicated validation of GOMOS profiles in the UTLS region has not been performed so far, although some results can be found in the extensive validation of GOMOS ozone profiles focused on the stratosphere. For IPF v.4.02, Meijer et al. (2004) reported a positive bias ~20% in tropical UTLS in comparisons with ozonesondes. In comparisons with ozonesondes at two polar stations, Tamminen et al. (2006) found ~10 % bias for Sodankylä (67.4°N, 23.6°E) and over 20% negative bias for Marambio (64.3°S, 56.7°W), for altitudes below 15 km. For GOMOS ozone profiles processed with IPF v.5.0, van Gijsel et

al. (2010) have performed an analysis analogous to Meijer et al. (2004) but on a significantly larger dataset and found large GOMOS positive bias, over 40%, in the tropical UTLS and below. This conclusion is in full agreement with the validation work of Mze et al. (2010) using ozone soundings from eight SHADOZ stations. The validation studies performed so far (which are also focused mainly on the stratospheric ozone) indicate, however, the presence of large GOMOS ozone bias in UTLS also in IPF v.6 data. For example, Adams et al. (2014) reported over 20% positive GOMOS bias in UTLS in

comparisons with OSIRIS/Odin v.5.0 profiles, which are of good quality in the UTLS (e.g., Cooper et al., 2011)). Hubert et al. (2015) performed validation of GOMOS v.6 profiles with ozonesondes and reported a large positive ozone bias in the UTLS and the troposphere nearly everywhere except at polar high latitudes (only winter-time measurements are available for these locations). Large positive GOMOS ozone biases in UTLS are also observed in data agreement tables created for the HARMOZ dataset (Sofieva et al., 2013), which consists of user-friendly vertically gridded ozone profiles from 6 limb-

viewing instruments (GOMOS, MIPAS, SCIAMACHY, OSIRIS, SMR and ACE-FTS).

The operational GOMOS retrieval algorithm is optimized mainly for the stratosphere and the mesosphere (Kyrölä et al., 2010; Sofieva et al., 2010), but not for retrievals in the UTLS. This paper presents the results of studies aimed at optimization of GOMOS retrievals in the UTLS region. These studies have been performed in the framework of the European Space Agency (ESA) project ALGOM (GOMOS Level 2 evolution studies). The paper is organized as follows.

First, in Section 2, we present the validation of GOMOS IPF v.6 ozone profiles in the UTLS region using ozonesonde data, which has been performed in the framework of the ESA DRAGON-3 project. Section 3 is dedicated to the discussion on sensitivity of GOMOS retrievals in the UTLS and the reason for large positive bias of IPF v6 ozone data. Section 4 is dedicated to the description of the new aerosol-insensitive retrieval algorithm in the UTLS. In sections 5 and 6, extensive validation and geophysical assessment of the GOMOS ozone data processed with the FMI scientific processor is presented.

Summary (Sect. 7) concludes the paper.





## 2    Validation of V6 ozone profiles with NDACC ozonesonde data

### 2.1    Data and methodology

GOMOS IPF Version 6 night-time ozone profiles (with solar zenith angles at tangent point larger than 105°) from the full mission are used in this work. The GOMOS data have been screened for outliers according to the recommendations in the

5    readme   document   (http://earth.eo.esa.int/pcs/envisat/gomos/documentation/RMF_0117_GOM_NL__2P_Disclaimers.pdf).
For validation, we used ozonesonde data from the Network for the Detection of Atmospheric Composition Change, NDACC (www.ndsc.ncep.noaa.gov). Figure 1 shows locations of the ozonesonde stations included in the comparison. Data availability is larger in the Northern Hemisphere (NH). We selected GOMOS and ozonesonde data separated less than 1000 km in ground distance, less than 3° in latitude and less than 24 hours in time. The information about the collocated GOMOS

10   and ozonesonde profiles in UTLS and in the troposphere is collected in Table 1.

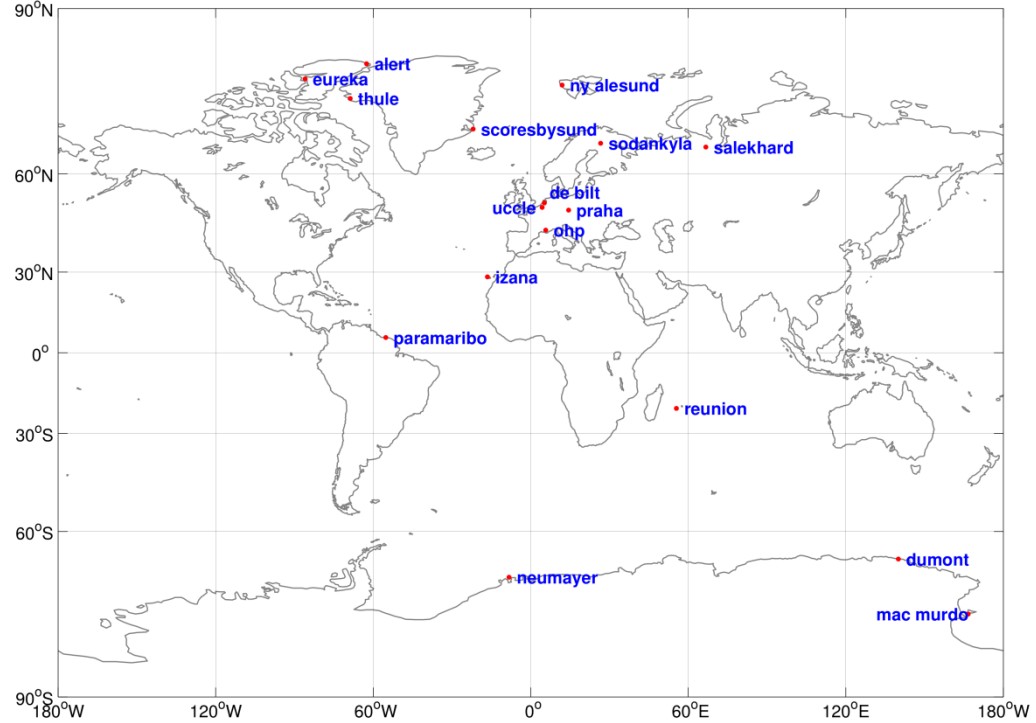

**Figure 1.  Locations of the NDACC ozonesonde stations considered in the comparison.**





**Table 1. Number of useful pairs of collocated GOMOS profiles and ozonesondes in UTLS and troposphere for the NDACC stations included in the comparison.**

| Station (Lat °N, Lon °E) | Number of collocations in UTLS | Number of collocations in the troposphere |
|---|---|---|
| Alert (82.45, -62.51) | 38 | 3 |
| De Bilt (52.10, 5.18) | 53 | 3 |
| Dumont (-66.67, 140.02) | 57 | 9 |
| Eureka (79.99, -85.93) | 51 | 3 |
| Mac Murdo (-77.85, 166.63) | 7 | 2 |
| Izaña (28.30, -16.50) | 160 | 67 |
| Neumayer (-70.68, -8.26) | 74 | 18 |
| Ny Alesund (78.93, 11.93) | 132 | 26 |
| OHP (43.94, 5.71) | 74 | 3 |
| Paramaribo (5.8, -55.22) | 75 | 15 |
| Praha (50.00, 14.44) | 66 | 0 |
| Reunion (-21.06, 55.48) | 82 | 34 |
| Salekhard (66.50, 66.70) | 7 | 0 |
| Scorebysund (-70.68, -8.26) | 112 | 20 |
| Sodankylä (78.93, 11.93) | 58 | 0 |
| Thule (43.94, 5.71) | 16 | 1 |
| Uccle (50.80, 4.35) | 195 | 6 |

GOMOS ozone profiles are interpolated to 1 km vertical grid. Ozonesonde data have been smoothed down to the vertical
5   resolution of GOMOS ozone profiles and also interpolated to the same vertical grid. For every collocated pair of profile, the
tropopause height is determined based on the WMO lapse-rate tropopause definition (WMO, 1957). The sonde temperature
profiles are used for the tropopause detection. A detailed description of the tropopause height calculation can be found in e.g.
Sofieva et al. (2014b).

The relative differences $\delta$ between GOMOS ozone and the ozonesondes are computed as $\delta = (G-S)/S \cdot 100\%$, where
10   $G$ refers to GOMOS and $S$ to the ozonesonde ozone number density at each altitude.



## 2.2    Validation results and discussion

Figure 2 shows the relative difference between GOMOS V6 and ozonesonde profiles from all the stations presented in Figure 1. The color scale of the dots in Figure 2 indicates the GOMOS retrieval uncertainty (in %). When looking at the differences as a function of altitude (Figure 2, left), it can be noticed that the median relative difference is largest below 15

km and always smaller than 10%. This small median difference is the result of the compensation between large individual differences of opposite sign at different stations.

In order to separate the tropospheric and stratospheric components, the relative differences are plotted as a function of the altitude relative to the tropopause (Figure 2, center). Below the tropopause, a large ozone overestimation (the bias is larger than 30%) by GOMOS can be observed. Removing the GOMOS data with uncertainty estimate larger than 75% or 50%

(Figure 2, green and blue lines, respectively) does not reduce the large positive bias observed below the tropopause. In this case, the bias is even larger than in the case with all data included in the comparison (red lines in Figure 2). This is because most of the data with large uncertainty corresponds to large negative differences. The best accuracy in GOMOS retrievals can be achieved using the brightest stars (the smallest noise). When only the 12 brightest stars are taken into account (right panel in Figure 2), the difference between GOMOS and the ozonesondes becomes larger (bias up to ~100%). The result is

similar to the one obtained when the GOMOS data with large retrieval uncertainty are removed.

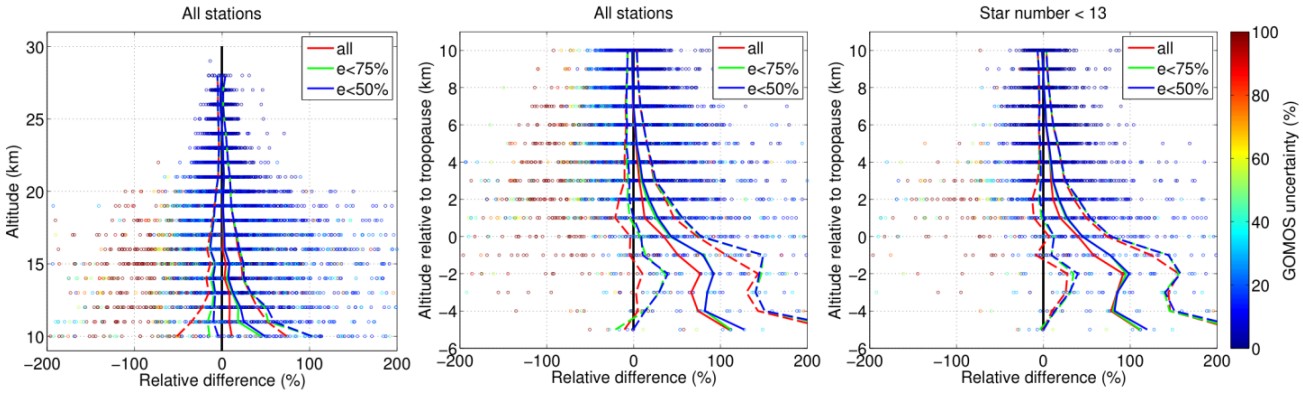

**Figure 2 Percentage relative difference (Eq. 1) from collocations with all NDACC stations as a function of altitude (left panel) and altitude relative to the tropopause (center) and for the 12 brightest stars (right). Colors of the points correspond to uncertainty estimates of the GOMOS ozone profiles. The lines indicate the median (solid) and the standard deviation (dashed) when the**

**GOMOS uncertainties are smaller than 50% (blue), 75% (green) and using all data (red).**

In order to evaluate the dependence of the biases with respect to ozonesonde profiles on latitude, a similar comparison on the altitude relative to the tropopause grid has been performed for different latitude regions (Figure 3). The largest overestimation of UTLS ozone by GOMOS is observed in the tropical region (Figure 3, bottom left), where also most of the data below the tropopause are available. The best agreement is found at high latitudes, especially for the stations in

Antarctica (Figure 3, bottom right), where the median RD value is smaller than 40%. At the NH mid-latitudes a negative bias can be observed below the tropopause and ~3 km above. No ground-based measurements are available at SH middle-




latitudes. These results are in perfect agreement with those reported in the recent paper by Hubert et al. (2015), who used a larger ozonesonde dataset for the validation. The strong overestimation of ozone in the tropical UTLS is observed also in comparisons with satellite measurements presented in (Sofieva et al., 2015) and in Section 6 of our paper. Note that many thousands of GOMOS occultations were used in comparison with satellites, thus making no doubts in statistical significance.

The presented comparison of GOMOS ozone profiles with ozonesonde data, together with other analyses overviewed in the introduction show a consistent picture: GOMOS V6 ozone is strongly overestimated in the UTLS (median relative difference up to 100%). The largest difference was observed in the tropics. The observed positive bias did not show any dependence on the star brightness, making impossible to select a subset of GOMOS dataset with realistic ozone profiles in the UTLS region. These validation results prompt us to develop an improved retrieval algorithm, which is discussed below.

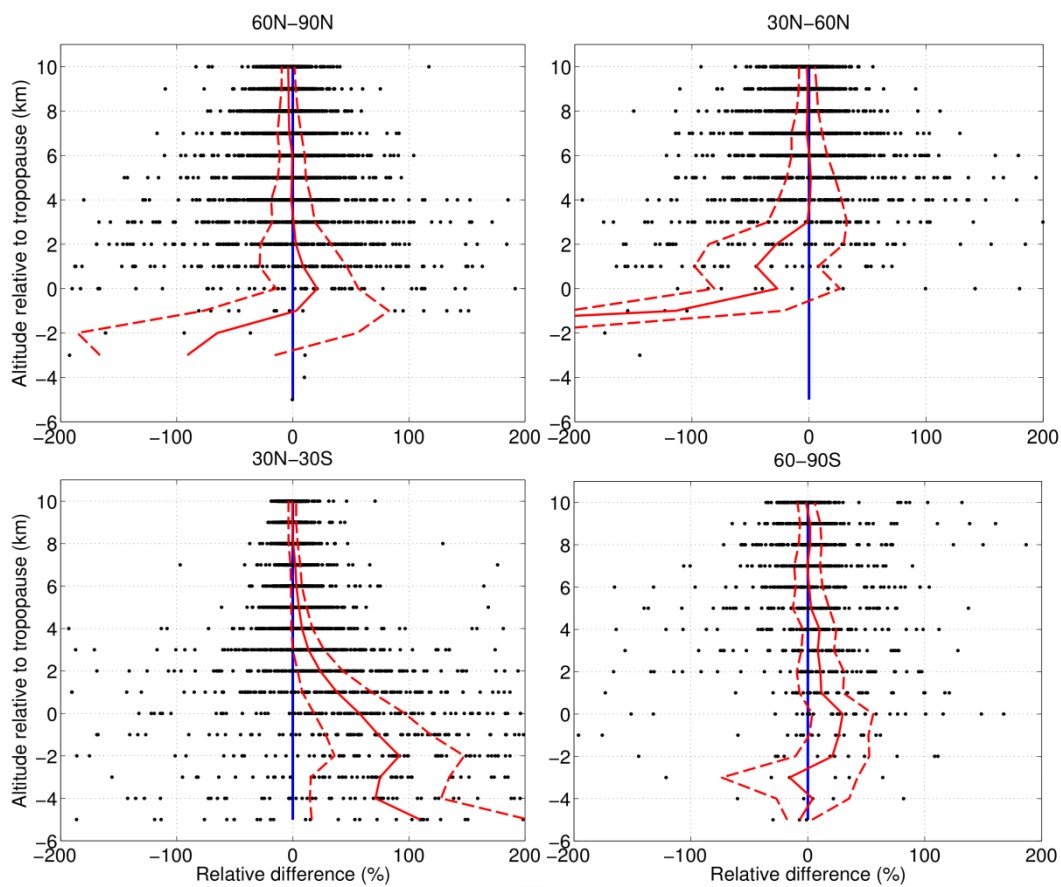

**Figure 3. Relative difference of ozone profiles at different latitude bands (indicated in the title of each subplot). The red lines indicate the median (solid) and the standard deviation (dashed).**




## 3 Identification of reasons for GOMOS V6 ozone biases in the UTLS

The history of GOMOS algorithm development indicates that a possible reason for the large ozone bias in the IPF v6 is the parameterization of aerosol extinction. We performed different sensitivity analyses with two scientific processors developed at FMI. The first one, GOMLAB, which is analogous to the official processor IPF v6.0: it relies on the two-step inversion and has all specific features implemented in IPF v6. GOMLAB has been used in several studies on development of GOMOS retrievals, for example, in (Sofieva et al., 2010; Tamminen et al., 2010). In the second processor, which we refer hereafter to as the one-step retrieval algorithm, local density profiles are retrieved directly from transmission spectra, i.e. in one step (Hakkarainen et al., 2013). All performed analyses show that the retrieval of ozone, aerosols and $NO_2$ in the UTLS is very sensitive to the assumed model of aerosol extinction. This sensitivity was shown previously in (Tamminen et al., 2010) and illustrated also below in Figure 4 with examples of ozone, aerosols and $NO_2$ retrievals for one occultation of the very bright star Canopus R04078/S002 (10 Dec 2002, 30°N 15°W) collocated with ozone sounding at Izana (time difference 11 h 37 min, distance 223 km). In Figure 4, three aerosol extinction models are considered: the Ångström law $\propto \frac{1}{\lambda}$ (two-step retrievals with GOMLAB, green lines), $1^{st}$ degree polynomial model in $1/\lambda$ (both one-step and two-step inversion, blue dashed and solid lines), and the second degree polynomial model in $1/\lambda$ (two-step retrievals with IFP v6, black lines). On the left panel of Figure 4, the ozone profile measured by the collocated ozonesonde is shown by the red line. As observed in Figure 4 (left), the V6 ozone profile has a large positive bias compared to ozonesonde. This bias is dramatically reduced when low-degree aerosol models are used (linear or $1/\lambda$). The changes in the UTLS ozone are accompanied with the changes in aerosols and $NO_2$. The retrieval results for one-step and two-step inversion are similar: the one-step profiles are noisier because less smoothing is applied in the inversion. With a stronger regularization, the one-step and two-step profiles look very similar.



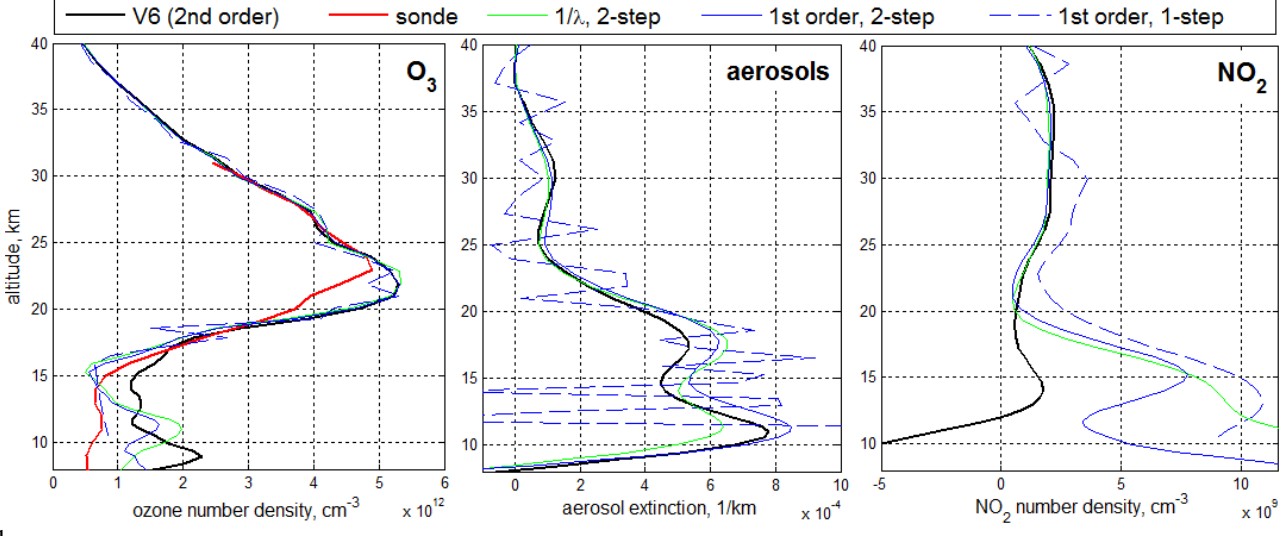

l

**Figure 4. Sensitivity of ozone (left), aerosols (center) and NO$_2$ (right) profile retrievals to the aerosol model used in inversion. The considered aerosol extinction models are ~1/λ model (green line), 1$^{st}$ degree polynomial in 1/λ (blue lines) and 2$^{nd}$ degree polynomial (V6, black line). The results for the 1$^{st}$ degree polynomial model are shown for both two-step and one-step inversion. The ozonesonde profile is shown by the red line in the left panel.**

The sensitivity of the GOMOS retrievals to the assumed aerosol model is large: ozone and aerosol number density can change nearly by a factor of 2, while changes in NO$_2$ can be up to a few hundreds of percent. Such sensitivity is due to several reasons. First, the signal-to noise ratio for stellar occultation measurements is relatively low in the UTLS. Second,

GOMOS retrievals in the UTLS use the visible wavelength 400-675 nm, where the overall shape of ozone, aerosols and NO$_2$ cross-sections can be approximated by a quadratic function (or a higher degree polynomial). Therefore, if the absolute cross-sections in the whole GOMOS VIS wavelength range are used, these constituents interfere in the retrievals. Third, a second degree polynomial has a general deficiency of for modelling aerosol extinction spectra: it gives more freedom to possible shapes of aerosol extinction spectra than can be observed for different composition of aerosol particles. Finally, we have to

note that GOMOS spectra have perturbations due to residual scintillations (Sofieva et al., 2009), uncertainty of the dilution correction and estimated extinction due to Rayleigh scattering. All these factors contribute to high sensitivity of GOMOS retrievals in the UTLS to the assumed model for aerosol extinction spectra.

This prompted us to develop a new aerosol-insensitive ozone retrieval algorithm in the UTLS. Using triplets in the Chappuis band - the method that is often used in retrievals from limb-scattering instruments (e.g., Degenstein et al., 2009; Flittner et

al., 2000) - has allowed a simple and a robust inversion in the UTLS, which only assumes that the aerosol extinction is linear in a relatively narrow wavelength band. Flittner et al. (2000) studied in details the performance of this DOAS-type algorithm under different aerosol loads and types and found that using triplets in the Chappuis band for ozone retrievals has a very low



sensitivity to aerosols. Below we present the details of the application of the triplet method to GOMOS retrievals and extensive assessment of the retrieval results.

## 4    GOMOS ozone inversion optimized for the UTLS (ALGOM2s)

In this section, we describe a new GOMOS retrieval algorithm, which uses aerosol-insensitive inversion in the UTLS. The proposed ozone retrieval algorithm in the whole GOMOS altitude range from the upper troposphere to the lower thermosphere inversion consists of the following steps:

1. Retrievals of horizontal column density ozone profile in the UTLS using visible triplets;

2. Forming the resulting horizontal column density ozone profile in the whole GOMOS range by weighted combination of the V6 ozone profiles in the middle atmosphere and the new triplet ozone profile in the UTLS.

3. Performing the vertical inversion in the same way as in IPF V6.

Below we describe each step in detail. Hereafter, we refer the new processing method and the new GOMOS ozone dataset to as "ALGOM2s v1.0". For short, we refer IPF v6 as to "V6".

### 4.1    Triplet inversion in the UTLS

In order to reduce the sensitivity of ozone retrievals to aerosols, we use the triplet method. We use the same wavelengths as in the classical triplet method by (Flittner et al., 2000): the reference wavelengths near 525 and 675 nm and absorbing wavelengths near 600 nm. Since stars are relatively weak sources of light, several pixels are used for reference and absorbing wavelengths (Figure 5). Using differential optical depth allows nearly cancelling scintillation-dilution perturbations and a significant reduction of uncertainty due to Rayleigh scattering correction based on the ECMWF field. The Level 1b transmission spectra $T_{L1b}(\lambda)$ at a given tangent altitude can be expressed as:

$$T_{L1b}(\lambda) = T_{ext}(\lambda) \cdot T_{dil}(\lambda) \cdot T_{sc}(\lambda) , \tag{1}$$

where $T_{dil}(\lambda)$ and $T_{sc}(\lambda)$ are transmittances due to dilution and scintillation, respectively.





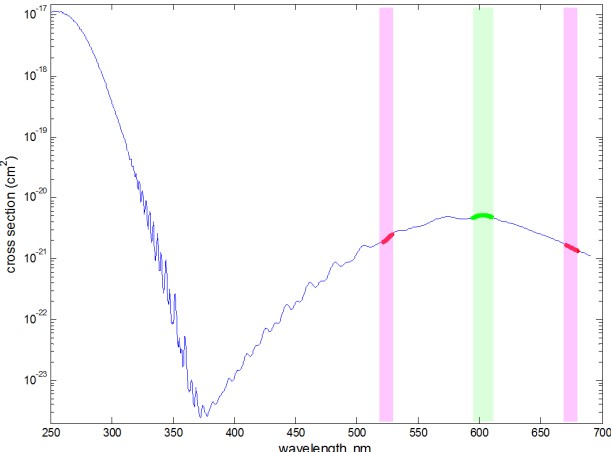

**Figure 5. Reference (red, 521-529 nm and 670-680 nm) and absorbing (green, 592-612 nm) wavelengths.**

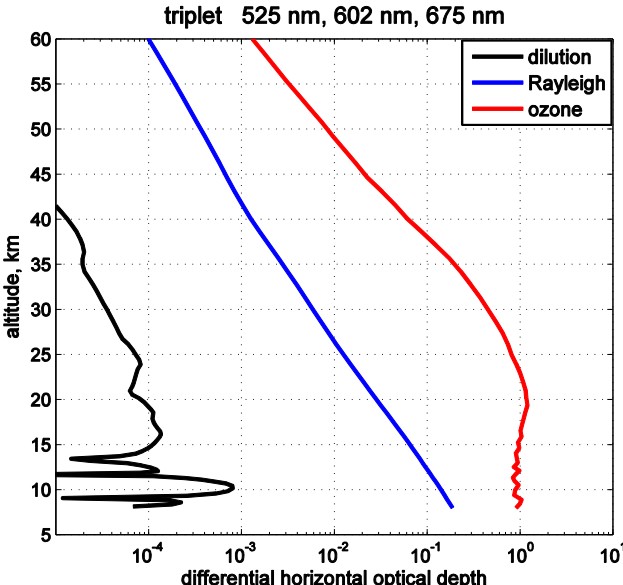

**Figure 6. Differential horizontal optical depth for the visible triplet 525 nm, 602 nm, 675 nm (see text for definition) due to ozone absorption, Rayleigh scattering and dilution. The contributions are computed based on the occultation R04078/S002.**

For each altitude, the differential optical depth $d\tau$ is computed as:

$$d\tau = \tau(\lambda_{ab}) - \frac{1}{2} \cdot \left( \langle \tau(\lambda_{r1}) \rangle + \langle \tau(\lambda_{r2}) \rangle \right) \tag{2}$$

where $\tau = -\ln(T_{L1b})$ is the optical depth, $\lambda_{ab}$ is absorbing wavelength, $\lambda_{r1}$ and $\lambda_{r2}$ are reference wavelengths. It has

contributions due to ozone absorption, Rayleigh and aerosol scattering, and due to refractive effects. However, due to the





selected wavelengths, ozone contribution to the differential optical depth strongly dominates. This is illustrated in Figure 6, which compares the contributions of ozone absorption, Rayleigh scattering and dilution (refractive attenuation) to the differential horizontal optical depth for the visible triplet 525 nm, 602 nm, 675 nm. This means that Level 1b transmittances, without dilution and scintillation correction, can be used for the inversion from the visible triplets. The contribution from

5 Rayleigh scattering to the differential optical depth is significantly smaller than that of ozone, but it can be as large as a few percent in the UTLS. Therefore, the Rayleigh optical depth is estimated (using the ECMWF field) and subtracted from the total optical depth data.

Computation of optical depth (taking logarithm) requires good signal-to-noise ratio. Only the pixels with the signal-to-noise ratio larger than 3 are used in the inversion. The uncertainty of the optical depth $\sigma_\tau$ is approximated as:

$$\sigma_\tau = \frac{\sigma_T}{T},$$
(3)

where $\sigma_T$ is uncertainty of transmittance $T$ (provided $\sigma_T \ll T$ ).

The retrievals of ozone line density from Level 1b data using visible triplet are performed only in the UTLS, i.e., at altitudes below $z_t$+7 km, $z_t$ is the tropopause height.

The average optical depth for reference channels $\langle\tau(\lambda_{r1})\rangle$ and $\langle\tau(\lambda_{r2})\rangle$ is used, and the triplet optical depth is computed

15 for each channel at absorbing wavelengths from 592 nm to 612 nm according to Eq.(2). The uncertainties of the differential optical depth values (Eq.(2)) are computed as:

$$\sigma_{d\tau}^2 = \sigma_\tau^2 + \frac{1}{4}\sigma_{r1}^2 + \frac{1}{4}\sigma_{r2}^2,$$
(4)

where $\sigma_\tau$ is uncertainty in the absorbing channel, $\sigma_{r1}$ and $\sigma_{r2}$ are uncertainties of the average optical depth in the reference channels.

20 Ozone line density is estimated for each pixel in absorbing channels $O_3(\lambda) = d\tau(\lambda) / D_{cross}(\lambda)$, where $D_{cross}(\lambda)$ is the differential cross-sections corresponding to a triplet, and the weighted mean of these estimates $\overline{O}_3$ is computed (with weights inversely proportional to uncertainties $\sigma_{O3}^2(\lambda_i)$ for individual absorbing channels). This is equivalent to the weighted least squared estimates using all triplets. The associated uncertainty of the triplet line density $\overline{O}_3$ is given by the uncertainty of the weighted mean:

$$\sigma_{\overline{O}_3}^2 = \frac{1}{\sum_{i=1}^{N} 1/\sigma_{O3}^2(\lambda_i)}$$
(5)





provided that there is no systematic uncertainty in different values of $O_3(\lambda_i)$. This assumption might be violated for the cases when the scintillation correction is not perfect. To take into account possible systematic uncertainties, the uncertainty of the triplet line density $\bar{O}_3$ is estimated as:

$$\sigma_{\bar{O}_3,corr}^2 = \frac{1}{\sum\limits_{i=1}^{N} 1/\sigma_{O3}^2(\lambda_i)} \cdot \frac{1}{(N-1)} \sum_{i=1}^{N} \frac{(O_3(\lambda_i)-\bar{O}_3)^2}{\sigma_{O3}^2(\lambda_i)} \tag{6}$$

5   In Eq. (6), the first factor is the uncertainty of the weighted mean, Eq.(5). The second factor in Eq. (6) takes into account variability between the different values of $O_3(\lambda_i)$. For some samples, it may turn that $\sigma_{\bar{O}_3,corr}^2 < \sigma_{\bar{O}_3}^2$; in this case, $\sigma_{\bar{O}_3}^2$ is taken as the uncertainty of the triplet line density.

## 4.2   Combining V6 and triplet horizontal column densities

The combining V6 and triplet ozone profiles is performed in the UTLS (from 6 km above the tropopause until the lowermost

10   altitude of an occultation).

The uncertainty of V6 is modified by adding a function increasing linearly from 0%  at ($z_t$+6 km) to 20% at the tropopause height $z_t$, with the saturation level of 20% below $z_t$ (Figure 7, left, labelled as "V6 systematic"). Such modification is assumed to characterize the systematic uncertainty of V6 ozone line density due to uncertainty of the aerosol model. The combined ozone profile is the weighted mean of the V6 and triplet line density profiles with the weights inversely

15   proportional to uncertainties: modified V6 uncertainty for V6 profile and $\sigma_{\bar{O}_3}$ (Eq.(6)) for the triplet inversion (Figure 7, center). As a result, above ($z_t$ + 6 km), ozone profile follows exactly V6 data. Below ($z_t$ + 6 km), the profiles are closer to the triplet inversion and practically coincide with the triplet inversion at the tropopause and below (Figure 7, right).



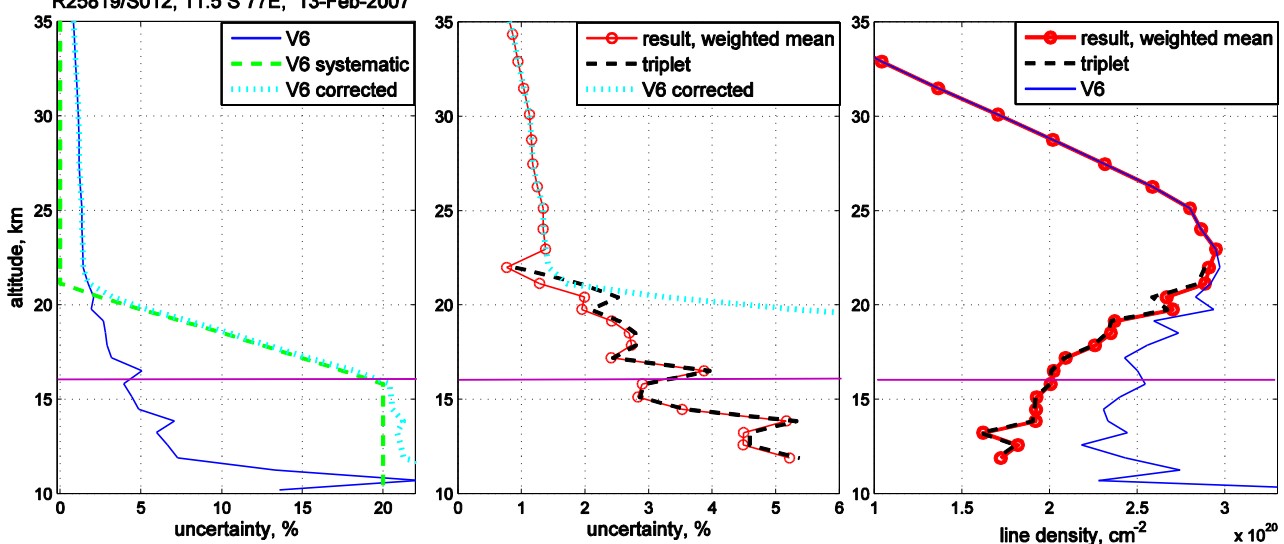

**Figure 7. Illustration of combining triplet and V6 line density profiles using the data from occultation R25819/S012 (11.5°S 77°E, 13 February 2007). Left: Original uncertainty of V6 ozone line density (blue line), systematic uncertainty (green), which is added quadratically to the original V6 uncertainty, and the resulting corrected V6 uncertainty (cyan). Center: V6 corrected uncertainty (cyan, as in the left panel), the uncertainty of the triplet inversion (black) and the resulting uncertainty of the weighted mean of V6 and triplet profiles (red). Right: V6 ozone line density (blue), ozone line density from triplet inversion (black), and the weighted mean of V6 and triplet profiles (red). The lapse-rate tropopause height is ~16 km is indicated my magenta horizontal line.**

If the lowest GOMOS altitude is above the tropopause, the triplet inversion is not performed and the profiles follow V6 data.

After combining the ozone line density profiles, the vertical inversion is performed in the same way as for IPF v6.

## 4.3    Examples of individual retrievals

To highlight the changes in retrieval results compared to IPF v6, the retrieved profiles for two occultations R09303/S002 and R04078/S002 are shown in Figure 8. For comparison, collocated ozonesonde profiles are also shown in Figure 8. As observed in Figure 8, the retrieved ozone profiles with the aerosol-insensitive method are much closer to the ozonesonde profiles than those of V6.

It is worth to note that the results of ALGOM2s retrievals are very stable with respect to some variations in reference and absorbing wavelengths used in the triplet inversion.





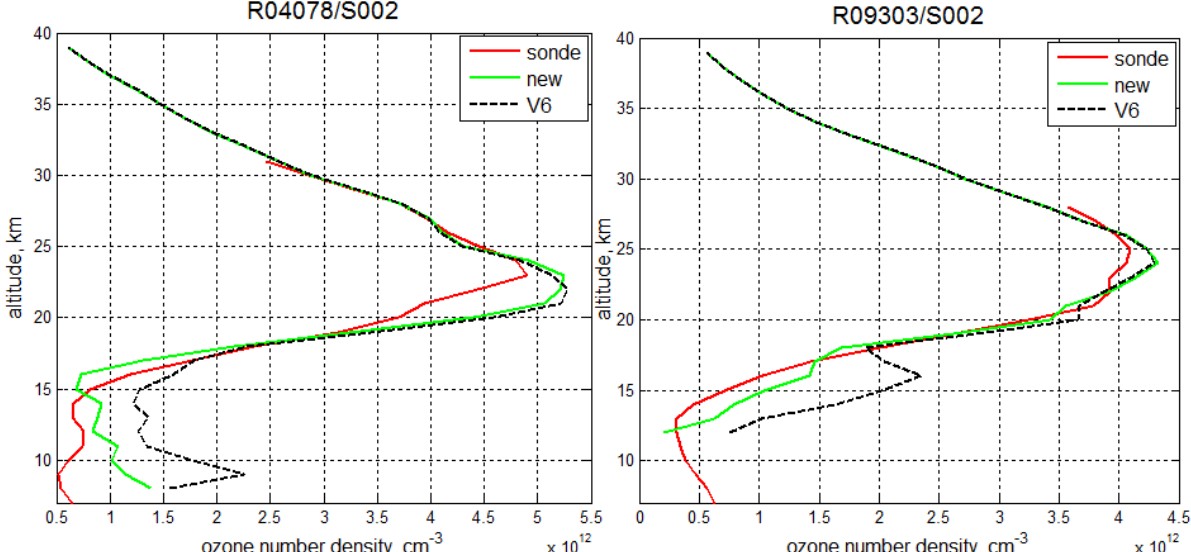

**Figure 8. Ozone profiles for occultations R09303/S002 (left) and R04078/S002(right) for the aerosol-insensitive retrievals ALGOM2s ("new") compared to V6 and ozone sonde profiles at Izana.**

## 5    Validation and intercomparison of the new retrievals: focus on the UTLS

### 5.1    Validation against NDACC ozonesondes

For validation against the NDACC ozonesondes, we have used the same data as in validation of V6 dataset presented in Sect.2. We show the results of comparisons for 50 brightest stars in GOMOS catalogue, as occultations of dim stars do not go usually below the tropopause.

Figures 9 and 10 show the results of the validation of ALGOM2s ozone profiles against NDACC ozonesondes for tropical and high-latitude stations, respectively. For comparison, the results for V6 ozone profiles are also shown. For tropical stations, the dramatic reduction of biases is observed. The ALGOM2s profiles are nearly unbiased with respect to ozonesonde data. At polar stations, the ALGOM2s results are not worse than those of V6 (at polar stations, also V6 data have a small bias in the UTLS in comparisons with ozonesondes). Also reduction of the spread in the UTLS is clearly observed for new ALGOM2s retrievals, as illustrated on the right panels of Figures 9 and 10.




**Figure 9. Statistics of comparison with ozonesondes at Izana (top), la Reunion (center) and Paramaribo (bottom). Left: median profiles (solid lines) and 16th and 84th percentiles (dotted lines). Right: solid lines: median of relative differences, dotted lines: 16th and 84th percentiles, dashed lines: the standard error of the mean.**





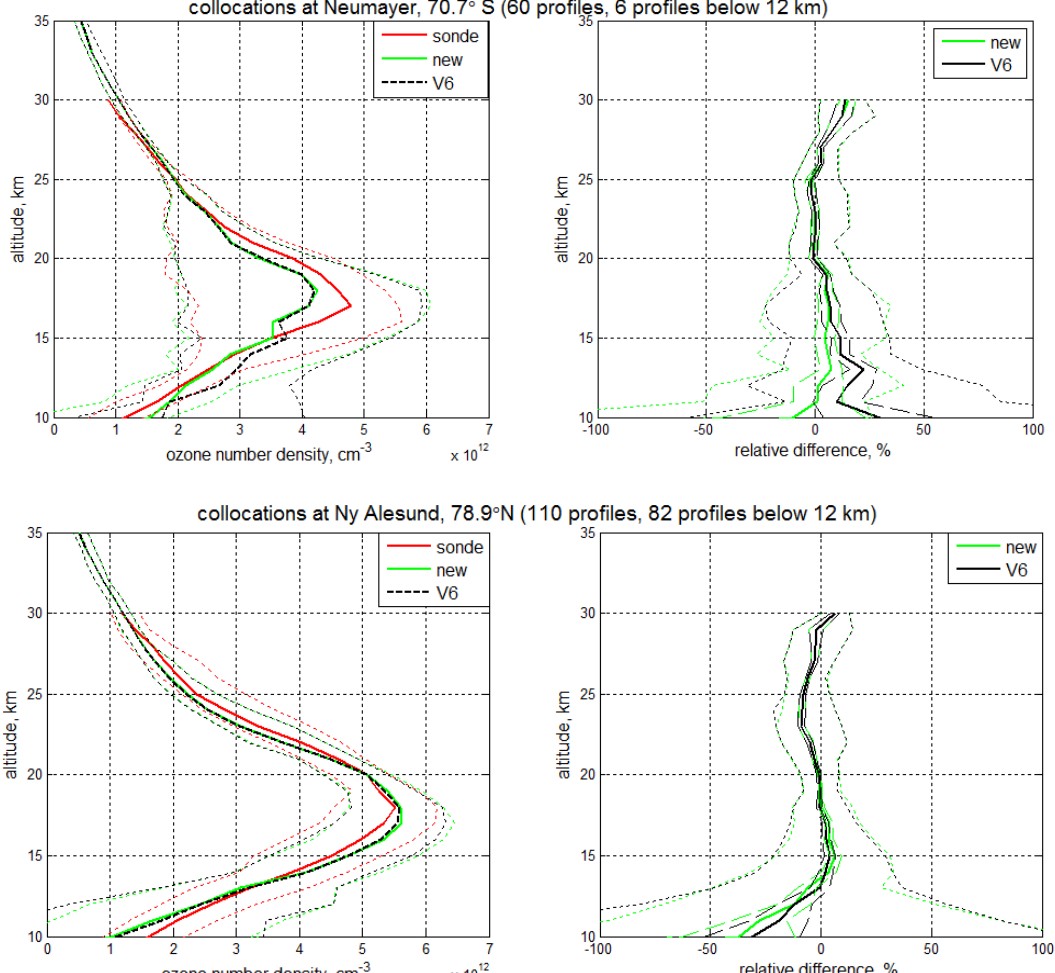

**Figure 10. As Figure 9, but for collocations at Neumayer (top) and Ny Alesund (bottom).**

## 6    Some geophysical illustrations in the UTLS using the ALGOM2s data

5    All the illustrations shown in the section are based on GOMOS dark-limb occultations (with solar zenith angle at tangent

point larger than 105°), which were processed with the ALGOM2s algorithm. The collection of ozone data is the same as

used in the Ozone_cci HARMOZ dataset (Sofieva et al., 2013) with small changes of removal of occultations of a few stars

with insufficient UV flux, which were not screened in the HARMOZ dataset. The applied screening of the GOMOS stars is

based on UV signal-to-noise ratio, thus it filters automatically the occultations, which might affected significantly by the

10    dark charge. The processed ozone data were screened for outliers according to recommendations written in GOMOS IPF 6.0

Disclaimer (outliers profiles are usually due to contamination by cosmic rays).



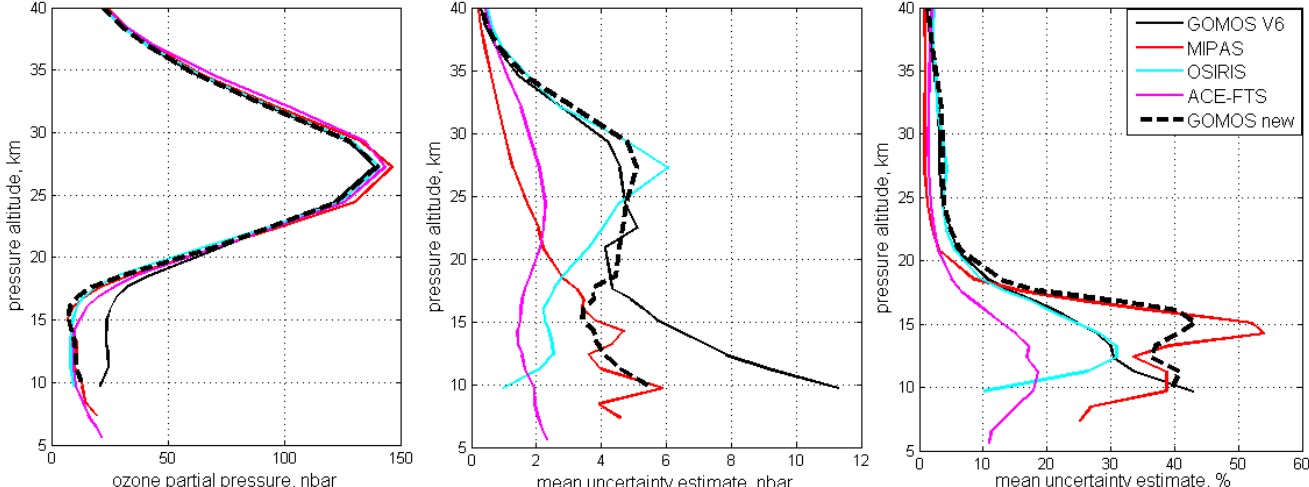

**Figure 11.** Left: mean ozone profiles at 20°S-20°N in 2007-2008 for GOMOS V6, MIPAS, OSIRIS, ACE-FTS and the aerosol-insensitive ALGOM2s GOMOS processor ('GOMOS new'). Center: mean ozone uncertainty estimates for each instrument in nbar. Right: mean ozone uncertainties in %.

As an illustration of a larger dataset, the GOMOS ozone profiles in the equatorial region (latitudes 20°S-20°N) in 2007-2008 are compared with ozone profiles from other satellite instruments -MIPAS (von Clarmann et al. 2003, 2009), OSIRIS (Degenstein et al., 2009) and ACE-FTS (Bernath et al., 2005) - which have been already used in several scientific studies in the UTLS (Aschmann et al., 2014; von Clarmann et al., 2007; Cooper et al., 2013; Kunze et al., 2010; Liu et al., 2009; Manney et al., 2011; Sioris et al., 2014). As observed in Figure 11 (left), the GOMOS V6 ozone profiles are strongly biased

in the tropical UTLS, while new GOMOS data are close to the profiles from other satellite instruments. A very good agreement of GOMOS and OSIRIS profiles in the middle stratosphere can be noticed on the left panel of Figure 11. For the aerosol-insensitive retrievals, the mean profile is close to that of OSIRIS also in the UTLS. The mean uncertainty of ALGOM2s ozone profiles are smaller in the UTLS than that of V6 data in absolute values (Figure 11, center), but the relative uncertainty of ALGOM2s ozone is larger in the UTLS than of V6 (Figure 11, right; note that the ALGOM2s ozone

concentration in the ITLS are twice smaller than that of V6).

    Seasonal variations and temporal evolution of ozone profiles in the UTLS is the subject of intensive research, e.g. (Gettelman et al., 2010; Hegglin et al., 2010; Randel and Jensen, 2013; Sioris et al., 2014). Figure 12 shows the temporal evolution of ozone profiles in the equatorial region (20°S-20°N) from OSIRIS and two versions of GOMOS data. The pronounced seasonal cycle associated with the variations in the tropopause height are clearly observed in all datasets. The

ozone values in the troposphere for the aerosol-insensitive retrievals are closer to those of OSIRIS than V6 profiles. It should be noted, that the coverage of the UTLS region by GOMOS data is limited due to applied screening by signal-to-noise ratio, thus the seasonal cycle is reproduced in GOMOS data with a significant sampling uncertainty.




The Asian Summer Monsoon (ASM) contains a strong anti-cyclonic vortex in the UTLS, spanning from Asia to the Middle East. The ASM has been recognized as a significant transport pathway for water vapor and pollutants to the stratosphere (e.g., (Kunze et al., 2010; Park et al., 2007)). Figure 13 shows ozone distributions at 100 hPa in June-August from OSIRIS, ACE-FTS, MIPAS, SCIAMACHY and GOMOS measurements. To obtain these maps, all available data have been used. For GOMOS, the maps are shown for both V6 and ALGOM2s ("GOMOS new") retrievals. The low ozone values in Asia associated with the strong upward motion of tropospheric air are clearly seen in these distributions, and peculiar features of ozone associated with the ASM are very similar in all datasets displayed. For V6, the ozone UTLS data have a significant positive offset, but lower values associated with ASM are observed in V6 distribution as well. For new GOMOS retrievals, the distribution is very similar to those by other satellite instruments.

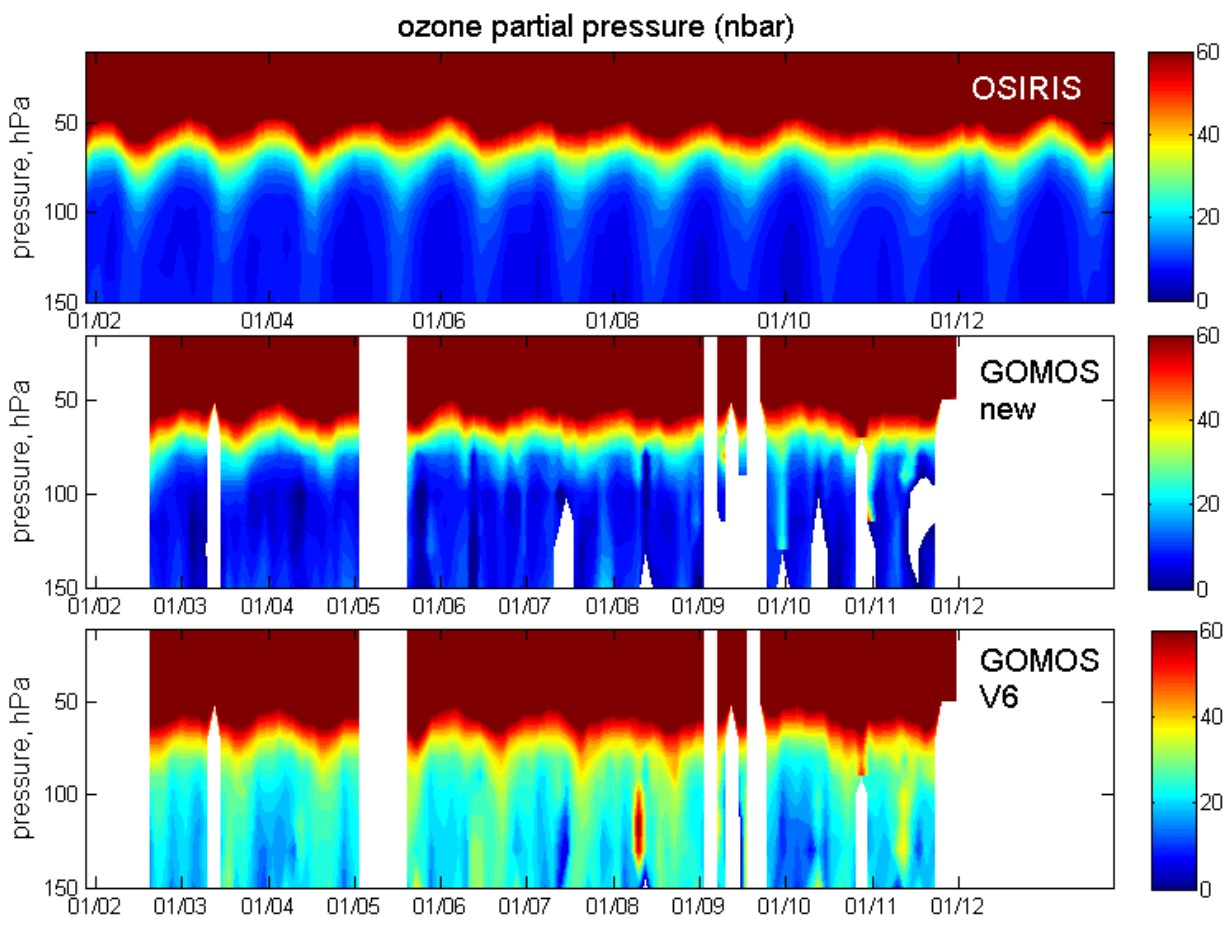

**Figure 12. Time series of ozone partial pressure profiles in nbar at 20°S–20°N from OSIRIS (top), GOMOS aerosol-insensitive retrievals (center) and GOMOS V6 (bottom).**




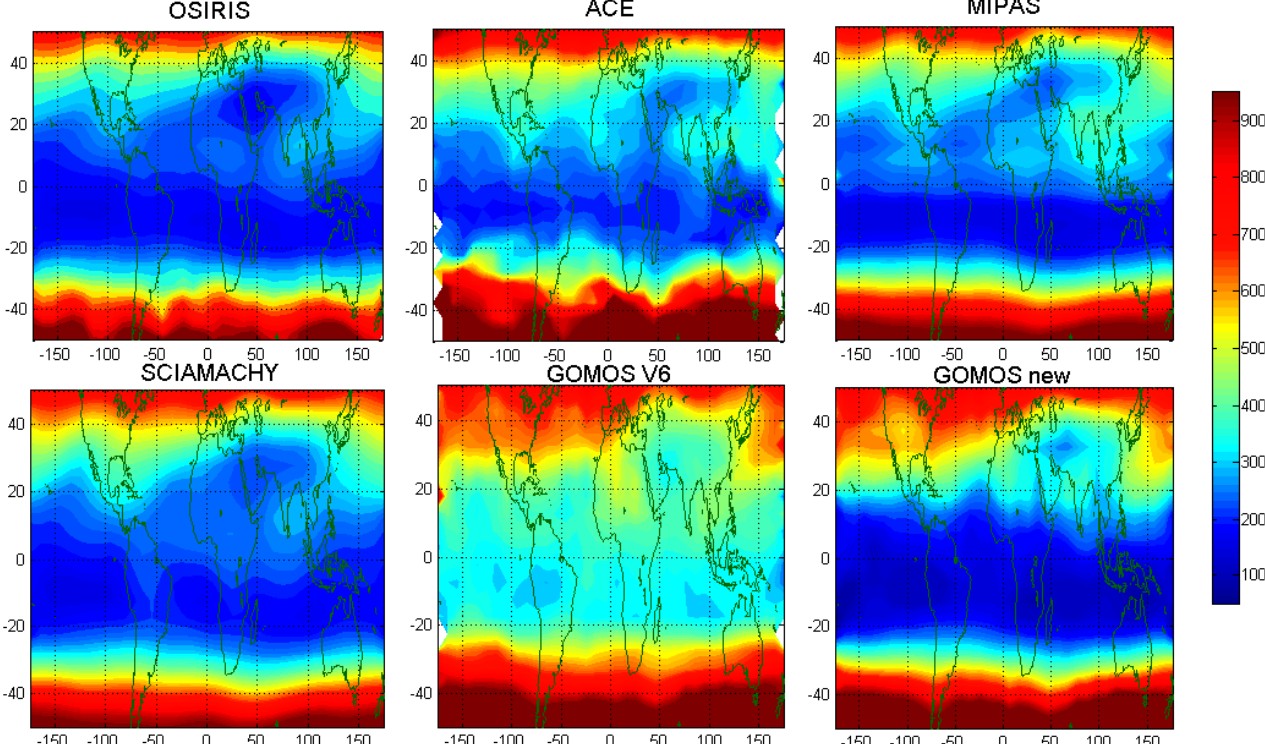

**Figure 13. Mean ozone mixing ratio (ppb) at 100 hPa in the summer season (June-August), as inferred from all available measurements by OSIRIS, ACE-FTS, MIPAS, SCIAMACHY and GOMOS (V6 and aerosol-insensitive retrievals).**

## 7    Summary and discussion

The satellite data with a good quality in the UTLS are very important for the studies of the complex processes and long-term changes in the UTLS. The validation of GOMOS IPF V6 ozone profiles against NDACC ozonesonde data has shown a strong overestimation (median relative difference up to 100%) of the ozone values by GOMOS IPF v6 in the UTLS. The largest difference was observed in the tropics. The observed positive bias does not show any dependence on the star brightness, thus making impossible to select a subset of GOMOS ozone V6 data with realistic ozone profiles in the UTLS region.

The extensive sensitivity analyses have shown that the current GOMOS retrievals of ozone, aerosols and NO2 are very sensitive to parameterization of aerosol extinction used in the inversion. In this paper, we have presented a new aerosol-insensitive algorithm for GOMOS ozone retrievals in the UTLS by using the visible triplets.

Using relatively narrow wavelength bands in the triplet inversion allows avoiding the scintillation and dilution correction (allows using Level 1b transmittances) and reducing uncertainty related to aerosol extinction spectral dependence





and removal of Rayleigh scattering. The proposed inversion using visible triplets in the UTLS is stable with respect to small variations of reference and absorbing wavelengths.

For the proposed inversion (ALGOM2s), the ozone profile follow V6 data in the middle atmosphere and follow the triplet ozone profiles in the UTLS and the troposphere. Such an approach seems to be advantageous for the ongoing

Ozone_cci project, http://www.esa-ozone-cci.org/. V6 ozone data have been extensively validated; they exhibit good quality and small biases with respect to ground-based measurements in the stratosphere (Adams et al., 2014; van Gijsel et al., 2010; Hubert et al., 2015; Laeng et al., 2014; Rahpoe et al., 2015). The ALGOM2s ozone data preserve all positive features of V6 in the stratosphere and have a significantly improved quality in the UTLS.

Validation of the new retrieved ozone profiles with ozone sondes and their comparison with V6 data has shown

dramatic reduction of ozone biases in the UTLS, especially in the tropics. The ALGOM2s ozone profiles are in good agreement with ozone sondes and other satellite data having a good quality in the UTLS. The geophysical phenomena are seen by the GOMOS dataset, but the coverage of the UTLS data by GOMOS is limited.

In the current dataset, all occultation of stars with insufficient UV flux (cool or/and dim stars) are excluded due to their problems in the upper atmosphere and also in the UTLS due to the influence of dark current noise, which increases during

the GOMOS mission (Tamminen et al., 2010). For climatologic studies, the application of aerosol-insensitive ozone inversion to averaged GOMOS transmittances, the approach used in (Fussen et al., 2004, 2005, 2010; Tétard et al., 2009), might provide a richer GOMOS climatologic dataset in the UTLS due to inclusion of more occultations and a less strict data filtering according to signal-to-noise ratio.

The ALGOM2s ozone dataset will be delivered in the user-friendly Ozone_cci netcdf format (Sofieva et al., 2013), on

both pressure and altitude grid; it is expected to replace the current GOMOS dataset used in the Ozone_cci project. It will be available at http://www.esa-ozone-cci.org/?q=node/161.

**Acknowledgements**

The development of the GOMOS algorithms has been performed in the framework of the ESA project ALGOM. The

validation against ozone sondes is supported by the ESA DRAGON-3 program. The comparison with satellite measurements has been performed in the framework of the ESA Ozone_cci project. The FMI team acknowledges the support by the Academy of Finland (project INQUIRE). The correlative data from balloon-based ozonesonde used in this publication were obtained from the NDACC network (www.ndsc.ncep.noaa.gov). The authors thank OSIRIS, MIPAS and ACE-FTS teams for providing the data.

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
