# Peer review of "Improved GOMOS/Envisat ozone retrievals in the upper troposphere and the lower stratosphere"

_Atmospheric Measurement Techniques, 2016_

## Referee Comment (RC1) · Anonymous Referee #1 · 19 Sep 2016

This paper presents a new retrieval algorithm for ozone profiles from the GOMOS instrument. The currently available data product, version 6, has a clear positive bias in the UTLS region that appears to be a result of interference from aerosol. This new ALGOMs v1.0 data product uses a well demonstrated DOAS-like retrieval technique to retrieve ozone in such a way as to be less sensitive to aerosol interference. The resulting data product appears to significantly reduce the bias in the UTLS in much better agreement with sondes and other satellite data sets. As always, it is important to publish updates to and validation of retrieval algorithms of widely used data sets and thus this paper is suitable for publication in Atmospheric Measurement Techniques. However, I do have a few questions/concerns regarding this paper and the work therein.

[Figure]

**Major Corrections:**

The use of the triplet method to produce ALGOMs v1.0 data is a good idea to mitigate the influence of aerosol in the UTLS. However, the methodology ought to be valid as a means of retrieval for the entire profile as well and, as such, would present an entirely new data product. The authors should present the fidelity of this data product across the full range of altitudes and compare to the V6 method. This is particularly applicable as the authors appear to choose an altitude cutoff ($Z_{TROP}$+6km) above which the triplet method is not included (or in the case where V6 terminates above the tropopause, the triplet method is not performed at all). Generally, arbitrary changes or transitions in the retrieval algorithm will cause anomalous effects that are revealed when analyzing data in bulk. While one would assume that the two methods would be nearly equivalent in the absence of aerosols higher in the atmosphere, I would guess that, due to algorithmic effects, they may not be. If the two methods do differ significantly at higher altitudes, my recommendation would be to release both as separate data products as well as a third, merged-data product (what the authors are currently presenting) as their recommended product to use. This would give maximum utility to the data user while presenting the opportunity for additional validation of the new retrieval. Of course, this recommendation of an extra data product is only a side-note to the authors and is not required for this paper.

Since the new ALGOMs methodology attempts to retrieve ozone in a way that is less sensitive to interference from aerosol, it would stand to reason that a better ozone retrieval should also result in a better aerosol retrieval. Since the retrieval of different species are often inter-dependent, it would be interesting to see how this new ozone retrieval impacts the resulting aerosol extinctions and whether they make sense or not when compared with other data sets. At the very least, the ALGOMs aerosol and V6 aerosol should be compared to ensure there are no sudden jumps in aerosol extinction at the transition, unless of course this new ozone retrieval has not been incorporated into a new aerosol extinction product as well. Then again, and at the discretion of the

editor, perhaps this is beyond the scope of this paper.

I do not agree with the use of the term "aerosol-insensitive" retrieval. While I would expect the triplet method to be much less sensitive to interference from aerosol loading, no sensitivity analysis is performed as was done to quantify the effect of different aerosol models on V6. This ties somewhat into the previous comment about needing to investigate the result on the aerosol product.

**Minor Corrections:**

Table 1 shows the number of colocations in the UTLS. How is this defined? Is it simply $Z_{TROP}$+6km as used later in the paper?

Page 7, Line 1 notes that "the results are in perfect agreement" with Hubert et al. 2015. "Perfect agreement" is too strong of a statement, particularly since no direct comparison is made between this study and Hubert et al. and cannot be made given the differences in scale of figures in that paper and this one. Additionally, Hubert et al. 2015 is now fully published and so the reference should be updated accordingly.

Earlier in the paper, the term ozone "horizontal column density" is used but then it appears this terminology is changed later in the paper to ozone "line density". To avoid any confusion related to spectroscopy, I would suggest maintaining the usage of the term "horizontal column density" throughout the paper. For the sake of brevity, simply introducing "HCD" may make things easier.

Page 15, Line 13: "Also reduction of the spread in the UTLS is clearly observed for new ALGOM2s retrievals, as illustrated on the right panels of Figures 9 and 10." With the exception of comparisons at La Reunion, I do not agree that a reduction is "clearly observed."

With regard to Figure 11 (right), normally I would advocate for showing percent comparisons over absolute comparisons. However, for this particular case, I would argue that 11 (right) does not add any informational value over 11 (center). Instead, given the
reduction in bias in the UTLS, it requires additional explanation to explain why the absolute uncertainty decreases and the relative uncertainty increases. As such, I would advocate eliminating the rightmost figure in Figure 11.

I would generally clean up the figures. At full size, they are legible but once they are shrunk down to a standard size for a published paper, many of the axes and labels will be too small to read.

**Grammatical Corrections:**

Page 9, Line 7: "ozone and aerosol number density" should be "ozone number density and aerosol extinction"

---

## Referee Comment (RC2) · A. Rozanov (Referee) · 30 Sep 2016

A. Rozanov (Referee)

alex@IUP.PHYSIK.UNI-BREMEN.DE

The manuscript describes an improved version of GOMOS/Envisat ozone retrieval algorithm resulted in a significant improvement of the retrieval quality in the UTLS region in comparison to the previous retrieval version (V6). The improved data quality achieved with the new retrieval technique is clearly of a great scientific importance for UTLS studies, where significant disagreement between the measurements from different satellite instruments still persists. I absolutely agree with the statement made by the authors in the beginning of summary section that "The satellite data with a good quality in the UTLS are very important for the studies of the complex processes and long-term changes in the UTLS". In this respect, however, the authors clearly miss

the point. While a great improvement with respect to the previous retrieval version is clearly demonstrated, no attempt is made to quantify the quality of the new retrieval. To my opinion, this issue significantly reduces scientific importance of the paper. All comparisons with satellite measurements are qualitative providing no possibility to the reader to estimate the quality of the new retrieval version. While the comparisons for the older version are made for a statistical ensemble of the sonde measurements, only a couple of examples are presented for the new retrieval version. Even for the previous version the sonde comparison is strongly biased to the middle and high northern latitudes including only one sonde station in the inner tropics, where the issues in UTLS seem to be strongest. For the reasons listed above I recommend a major revision of the manuscript to include a quantitative estimation of the quality of the new retrieval version in UTLS using a representative set of ozone sonde stations.

**Detailed comments**:

- Page 1, line 23 (and throughout the text): The notation "aerosol-insensitive" used with respect to the new retrieval version is confusing. Actually the authors mean that the new version does not depend on the aerosol parameterization used in the retrieval rather than the fact that the retrieval is insensitive to the presence of the aerosols in UTLS. To my knowledge, the experience of the University of Saskatchewan group with both OSIRIS and OMPS data shows that the triplet method is non-negligibly sensitive to the aerosol extinction. In this sense the notation "aerosol-insensitive" is wrong.

- Page 1, line 26: The notation "horizontal column ozone densities" is not common and thus should not be used in the abstract without any additional explanations.

- Page 1, line 27: The notation "triplet ozone profiles" is not common and thus should not be used in the abstract without any additional explanations.

- Section 2.1: The selection of sonde stations is too much biased to the middle

and high northern latitudes, only one station is used in the inner tropics, no stations are used in the southern mid-latitudes. Some additional stations need to be added in these regions.

- Fig. 3: It would be interesting to see these plots for absolute altitudes rather than only relative to the tropopause.

- Fig. 3: It should be discussed why median rather than more common mean values are plotted.

- Page 6, line 26: "No ground-based measurements are available at SH middle latitudes." - Why? There is a bunch of stations at these latitudes, e.g., Broadmeadows, Lauder, Macquarie Island, Ushuaia.

- Page 9, line 20: "the aerosol extinction is linear in a relatively narrow wavelength band" - I am not really sure that 525-675 nm band can be referred to as "narrow".

- Section 4: for a quick comparison it would be nice to know which wavelengths were used in V6 and if a differential or absolute radiance was used for a spectral fit. Furthermore, a rough idea of the vertical retrieval method (2-3 sentences) would be also helpful (even if it has been already discussed in details in previous publications).

- Fig. 5 is not really necessary.

- Page 12, line 12: What is "ozone line density", is it the same as "horizontal column ozone densities" used before?

- Page 13, Eq. (6): "N" is not defined

- Page 13, Eq. (6): To my opinion only the systematic uncertainties which are different between the individual absorbing channels can be accounted for in this
way. If you agree please include the corresponding remark in text. Otherwise please explain why the systematic uncertainties are accounted for by using this formula.

- Page 15, line 10: "For tropical stations, the dramatic reduction of biases is observed." - actually only Paramaribo is a real tropical station, the other two are already in a transition region. It is clearly seen in the comparisons showing clearly different results for Paramaribo in comparison with other two stations. As the improvements are strongest in the tropical region, the robustness of the conclusions would certainly benefit if more tropical stations are included.

- Caption of Fig. 9: please explain the meaning of "$16^{th}$ and $84^{th}$ percentiles"

- Fig. 11: Relative deviations between the satellite measurements need to be plotted to give a quantitative estimation for the quality of the new dataset.

- Page 19, line 3: Please provide approximate altitude for 100 hPa

- Fig. 12: An additional altitude grid should be provided or pressure grid should be replaced to have the same vertical axes as in previous plots.

- Fig. 12: The goal of the figure is not clear. Indeed, one sees a clear positive bias of V6 at lover altitudes. One also sees, however, that OSIRIS and new GOMOS data are still quite different. No further conclusions can be drawn from this plot. My suggestion is to extend/replace it by a couple of 2D plots for different latitudes showing both data sets as functions of time at a particular altitude level.

**Technical corrections**:

- Page 18, line 15: "ITLS" $\longrightarrow$ "UTLS"

- Page 20, line 12: "NO2" $\longrightarrow$ "$NO_2$"

---

## Author Comment (AC1) · 17 Nov 2016

Dear Reviewer,

Thank you very much for your valuable comments on our paper amt-2016-219 "Improved GOMOS/Envisat ozone retrievals in the upper troposphere and the lower stratosphere", which are taken into account in the revised version. In addition (as suggested by Reviewer#2), the enlarged significantly the ozonesonde dataset used for validation and updated the sections related to validation against ozonesonde profiles.

Below we present the detailed replies to your comments. Your comments are in blue, replies are in black font.

**Major corrections**

The use of the triplet method to produce ALGOMs v1.0 data is a good idea to mitigate the influence of aerosol in the UTLS. However, the methodology ought to be valid as a means of retrieval for the entire profile as well and, as such, would present an entirely new data product. The authors should present the fidelity of this data product across the full range of altitudes and compare to the V6 method. This is particularly applicable as the authors appear to choose an altitude cutoff ($Z_{TROP}$ +6km) above which the triplet method is not included (or in the case where V6 terminates above the tropopause, the triplet method is not performed at all).

Generally, arbitrary changes or transitions in the retrieval algorithm will cause anomalous effects that are revealed when analyzing data in bulk. While one would assume that the two methods would be nearly equivalent in the absence of aerosols higher in the atmosphere, I would guess that, due to algorithmic effects, they may not be. If the two methods do differ significantly at higher altitudes, my recommendation would be to release both as separate data products as well as a third, merged-data product (what the authors are currently presenting) as their recommended product to use. This would give maximum utility to the data user while presenting the opportunity for additional validation of the new retrieval. Of course, this recommendation of an extra data product is only a side-note to the authors and is not required for this paper.

In the middle and upper stratosphere, GOMOS ozone retrievals from the visible triplet only have very large uncertainties; they are much poorer than V6 ozone profiles (This is quite expected. The necessity of UV channels for good quality of ozone profiles has been discussed in many papers. This is especially important for stellar occultation measurements, due moderate signal-to-noise ratio). This is the reasons of combining the triplet inversion with V6 ozone profiles. We added a corresponding note on page 11 of the revised manuscript.

We are planning to release only the user-friendly ALGOM2s ozone dataset. Interested users can acquire all intermediate data products from the corresponding author. This note is added on page 21 of the revised version.

Since the new ALGOMs methodology attempts to retrieve ozone in a way that is less sensitive to interference from aerosol, it would stand to reason that a better ozone retrieval should also result in a better aerosol retrieval. Since the retrieval of different species are often inter-dependent, it would be interesting to see how this new ozone retrieval impacts the resulting aerosol extinctions and whether they make sense or not when compared with other data sets. At the very least, the ALGOMs aerosol and V6 aerosol should be compared to ensure there are no sudden jumps in aerosol extinction at the transition, unless of course this new ozone retrieval has not been incorporated into a new aerosol extinction product as well. Then again, and at the discretion of the editor, perhaps this is beyond the scope of this paper.

The ALGOM2s algorithm provides ozone only, no other species. You are right that improved ozone can be used for improved aerosol retrievals. We can speculate that one of the method can be removal of transmittances due to ozone, $NO_2$, and Rayleigh scattering from measured transmittances corrected for refractive attenuation and scintillation. We are planning such analyses in future. However, we have to note that retrievals of aerosol extinction (especially its wavelength dependence) from individual GOMOS occultations is a rather complicated, because if limited wavelength range (one might consider also using data from GOMOS infrared spectrometer, as e.g., in Vanhellemont, 2016, AMT), and perturbations of transmission spectra by incompletely corrected chromatic scintillation in case of oblique occultations. Other retrieval methods can be considered as well. All these can be the subject of future work.

I do not agree with the use of the term "aerosol-insensitive" retrieval. While I would expect the triplet method to be much less sensitive to interference from aerosol loading, no sensitivity analysis is performed as was done to quantify the effect of different aerosol models on V6. This ties somewhat into the previous comment about needing to investigate the result on the aerosol product.

In the revised version, the term "aerosol-insensitive" retrieval is not used.

**Minor corrections**
Table 1 shows the number of colocations in the UTLS. How is this defined? Is it simply ZTROP +6km as used later in the paper?

In the revised version, we extended the dataset used for validation, and presented the information about the number of collocated profiles as a latitude-altitude color plot (Figure 1, bottom of the revised manuscript).

Page 7, Line 1 notes that "the results are in perfect agreement" with Hubert et al. 2015. "Perfect agreement" is too strong of a statement, particularly since no direct comparison is made between this study and Hubert et al. and cannot be made given the differences in scale of figures in that paper and this one. Additionally, Hubert et al. 2015 is now fully published and so the reference should be updated accordingly.

We updated the reference to (Hubert et al., 2016) paper. In the revised version of the paper, we use exactly the same dataset as in (Hubert et al., 2016), therefore this note is removed.

Earlier in the paper, the term ozone "horizontal column density" is used but then it appears this terminology is changed later in the paper to ozone "line density". To avoid any confusion related to spectroscopy, I would suggest maintaining the usage of the term "horizontal column density" throughout the paper. For the sake of brevity, simply introducing "HCD" may make things easier.

The term "horizontal column density " is now used throughout the paper.

Page 15, Line 13: "Also reduction of the spread in the UTLS is clearly observed for new ALGOM2s retrievals, as illustrated on the right panels of Figures 9 and 10." With the exception of comparisons at La Reunion, I do not agree that a reduction is "clearly observed."

In the revised version, we included a figure (Fig. 12) illustrating the changes in the spread as a function of latitude and altitude relative to tropopause.

With regard to Figure 11 (right), normally I would advocate for showing percent comparisons over absolute comparisons. However, for this particular case, I would argue that 11 (right) does not add any informational value over 11 (center). Instead, given the reduction in bias in the UTLS, it requires additional explanation to explain why the absolute uncertainty decreases and the relative uncertainty increases. As such, I would advocate eliminating the rightmost figure in Figure 11.

The rightmost panel of the Figure 11 was removed and replaced with the panel showing relative difference between ozone profiles from the satellite measurements (suggested by Reviewer#2).

I would generally clean up the figures. At full size, they are legible but once they are shrunk down to a standard size for a published paper, many of the axes and labels will be too small to read.

We have improved the quality of the figures.

**Grammatical Corrections:**

Page 9, Line 7: "ozone and aerosol number density" should be "ozone number density and aerosol extinction"

Corrected.

---

## Author Comment (AC2) · 17 Nov 2016

Dear Alexey,

Thank you very much for your valuable comments on our paper amt-2016-219 "Improved GOMOS/Envisat ozone retrievals in the upper troposphere and the lower stratosphere", which are taken into account in the revised version.

Below we present the detailed replies to your comments. Your comments are in blue, replies are in black font.

**Major corrections**

The manuscript describes an improved version of GOMOS/Envisat ozone retrieval algorithm resulted in a significant improvement of the retrieval quality in the UTLS region in comparison to the previous retrieval version (V6). The improved data quality achieved with the new retrieval technique is clearly of a great scientific importance for UTLS studies, where significant disagreement between the measurements from different satellite instruments still persists. I absolutely agree with the statement made by the authors in the beginning of summary section that "The satellite data with a good quality in the UTLS are very important for the studies of the complex processes and long-term changes in the UTLS". In this respect, however, the authors clearly miss the point. While a great improvement with respect to the previous retrieval version is clearly demonstrated, no attempt is made to quantify the quality of the new retrieval. To my opinion, this issue significantly reduces scientific importance of the paper. All comparisons with satellite measurements are qualitative providing no possibility to the reader to estimate the quality of the new retrieval version. While the comparisons for the older version are made for a statistical ensemble of the sonde measurements, only a couple of examples are presented for the new retrieval version. Even for the previous version the sonde comparison is strongly biased to the middle and high northern latitudes including only one sonde station in the inner tropics, where the issues in UTLS seem to be strongest. For the reasons listed above I recommend a major revision of the manuscript to include a quantitative estimation of the quality of the new retrieval version in UTLS using a representative set of ozone sonde stations.

The objective of the original manuscript was introducing the new GOMOS retrieval method. Therefore, the validation in the original manuscript was not extensive as possible, but informative (to identify main problems and demonstrate improvements). The V6 biases in the tropical UTLS are large and evident, as well as improvements with the new algorithm.

However, we do agree that as extensive as possible validation would be definitely beneficial and would increase the scientific importance of the paper and the corresponding dataset. Therefore, we decided to use all available ozone sonde measurements from NDACC, WOUDC and SHADOZ network (the dataset, which is used in Ozone_cci project and in

(Hubert et al., 2016)). Daan Hubert (BIRA) kindly provided the collocated dataset of ozonesonde measurements and he is now the co-author of the paper.

In the revised manuscript, the text in the sections related to ozonesonde validation (Sect. 2.2 and 5.1) is modified, and the figures in Sect 2.2. are replaced by new ones, and 2 new figures are added to Sect. 5.1.

We also included more quantitative comparisons with satellite measurements, as suggested.

However, the results and conclusions are the same as in the original manuscript (as expected).

**Detailed comments**:

Page 1, line 23 (and throughout the text): The notation "aerosol-insensitive" used with respect to the new retrieval version is confusing. Actually the authors mean that the new version does not depend on the aerosol parameterization used in the retrieval rather than the fact that the retrieval is insensitive to the presence of the aerosols in UTLS. To my knowledge, the experience of the University of Saskatchewan group with both OSIRIS and OMPS data shows that the triplet method is non-negligibly sensitive to the aerosol extinction. In this sense the notation "aerosol-insensitive" is wrong.

In the revised version, the term "aerosol-insensitive" retrieval is not used.

Page 1, line 26: The notation "horizontal column ozone densities" is not common and thus should not be used in the abstract without any additional explanations.

Page 1, line 27: The notation "triplet ozone profiles" is not common and thus should not be used in the abstract without any additional explanations.

We have modified slightly the text in the abstract so that these specific terms are not used in the revised abstract.

Section 2.1: The selection of sonde stations is too much biased to the middle and high northern latitudes, only one station is used in the inner tropics, no stations are used in the southern mid-latitudes. Some additional stations need to be added in these regions.

Page 6, line 26: "No ground-based measurements are available at SH middle latitudes." - Why? There is a bunch of stations at these latitudes, e.g., Broadmeadows, Lauder, Macquarie Island, Ushuaia.

As explained above, now all available ozonesonde data are used for validation.

Fig. 3: It would be interesting to see these plots for absolute altitudes rather than only relative to the tropopause.

The figure is replaced with the global distribution of biases, as a function of latitude and altitude. The distributions are presented for both absolute and relative to tropopause altitudes.

Fig. 3: It should be discussed why median rather than more common mean values are plotted.

A note is added in the beginning of Section 2.2 of the revised version.

Page 9, line 20: "the aerosol extinction is linear in a relatively narrow wavelength band" - I am not really sure that 525-675 nm band can be referred to as "narrow".

In the content of the sentence, it is narrow compared to the whole UV-VIS wavelength range used in IPF V6 retrievals. To avoid misinterpreting, we changed "relatively narrow wavelength band" into "wavelength band of ~150 nm"

Section 4: for a quick comparison it would be nice to know which wavelengths were used in V6 and if a differential or absolute radiance was used for a spectral fit. Furthermore, a rough idea of the vertical retrieval method (2-3 sentences) would be also helpful (even if it has been already discussed in details in previous publications).

We include a paragraph with more details about the GOMOS IFP V6 spectral and vertical inversion, with corresponding references, into the introduction (Sect. 1). In section 4, we refer to this description.

Fig. 5 is not really necessary.

We would prefer keeping this figure, for illustration of position of the triplet wavelength and the relation to ozone cross-sections.

Page 12, line 12: What is "ozone line density", is it the same as "horizontal column ozone densities" used before?

Yes. The term "horizontal column density" is now used throughout the paper.

Page 13, Eq. (6): "N" is not defined

The variable is now explained.

Page 13, Eq. (6): To my opinion only the systematic uncertainties which are different between the individual absorbing channels can be accounted for in this way. If you agree please include the corresponding remark in text. Otherwise please explain why the systematic uncertainties are accounted for by using this formula.

Of course, Eq. (6) assumes that systematic uncertainties are different for absorbing channels (not a constant systematic bias). In the considered case, such uncertainties might result from incomplete chromatic scintillation correction. We included a corresponding remark in the revised version.

Page 15, line 10: "For tropical stations, the dramatic reduction of biases is observed." - actually only Paramaribo is a real tropical station, the other two are already in a transition region. It is clearly seen in the comparisons showing clearly different results for Paramaribo in comparison with other two stations. As the improvements are strongest in the tropical region, the robustness of the conclusions would certainly benefit if more tropical stations are included.

As explained above, now all available ozonesonde data are used for validation. The results and conclusions are the same.

Caption of Fig. 9: please explain the meaning of "16th and 84 th percentiles"

An explanation is added.

Fig. 11: Relative deviations between the satellite measurements need to be plotted to give a quantitative estimation for the quality of the new dataset.

The relative deviation between the satellite measurements are shown in the rightmost panel of the updated Fig.11 (Fig.13 in the revised manuscript). The previous rightmost panel has been removed according to the suggestion by Reviewer#1.

• Page 19, line 3: Please provide approximate altitude for 100 hPa

Added.

• Fig. 12: An additional altitude grid should be provided or pressure grid should be replaced to have the same vertical axes as in previous plots.

The altitude grid is also provided in the updated version of Fig. 12 (Fig.14 in the revised manuscript).

• Fig. 12: The goal of the figure is not clear. Indeed, one sees a clear positive bias of V6 at lower altitudes. One also sees, however, that OSIRIS and new GOMOS data are still quite different. No further conclusions can be drawn from this plot. My suggestion is to extend/replace it by a couple of 2D plots for different latitudes showing both data sets as functions of time at a particular altitude level.

Fig. 12 of the original manuscript was included in order to observe weather the seasonal cycle in tropical UTLS ozone is seen in GOMOS data. The presented figure answers fully this question. Additionally, it illustrates also the limited coverage of the UTLS by GOMOS. The presentation as 2D plots would be less visible, also because of very large bias of the V6 ozone data in the UTLS.

**Technical corrections:**
• Page 18, line 15: "ITLS" →"UTLS"

• Page 20, line 12: "NO2"→ "$NO_2$"

Corrected.